



**Reply to Comment on Franz et al. (2023): A reinterpretation of the 1.5 billion year old Volyn**
**'biota' of Ukraine, and discussion of the evolution of the eukaryotes, by Head et al. (2023)**
**Gerhard Franz[1], Vladimir Khomenko[1,2], Peter Lyckberg[3], Vsevolod Chornousenko[4], Ulrich**
**Struck[5]**
[1]Institut für Angewandte Geowissenschaften, Technische Universität Berlin, D-10587 Berlin,
Germany
[2]M.P. Semenenko Institute of Geochemistry, Mineralogy and Ore Formation, The National
Academy of Sciences of Ukraine, 34, Palladina av., Kyiv, 03142, Ukraine
[3]Luxembourg National Museum of Natural History, 25 Rue Münster, 2160 Luxembourg,
Luxembourg
[4]Volyn Quartz Samotsvety Company, Khoroshiv (Volodarsk-Volynski), Ukraine[*]
[5]Museum für Naturkunde, Leibniz-Institut für Evolutions- und Biodiversitätsforschung,
Invalidenstraße 43, Berlin, D-10115, Germany
[*]now at: Kondratyuka str. 9, al. 25, Zhytomyr, 10009 Ukraine
Correspondence: Gerhard Franz (gerhard.franz@tu-berlin.de; gefra548@gmail.com)

**Abstract**. Head et al. (2023) emphasize the importance of the Volyn biota for the evolution,
especially in the so-called 'boring billion', in a detailed outline about the biological and
geological context. However, they question that the Volyn biota represent Precambrian fossils
and instead argue that they are young contaminants of 'museum dust'. In addition, they
postulate that they are of a-biotic origin. We present here a detailed discussion of their points
of concern based on presented data, including some additional information. Their points of
concern were:
   - One object, shown by Franz et al. (2023) is similar to a pollen grain, another object is
     similar to trichomes; we show indications for fossilization and summarize our
     arguments against 'museum dust'.
   - They question the fossil character of the biota and argue for a biomineralization; we
     show that the biomineralization in trichomes is distinct from the mineralization of the
     biota.
   - They missed information about the internal structure; we repeat the presented
     information about the internal structure in more detail, which is also indicative of fossil
     material and inconsistent with trichomes.
   - They argue that we did not compare via infrared spectroscopy the biota with recent
     fungi; since the biota experienced temperatures near 300°C, we think that a
     comparison with thermally degraded chitosan is more appropriate.
   - They question the use of strongly negative $\delta^{13}$C as an argument for biotic origin, but
     we show that in combination with positive $\delta^{15}$N values and the geological situation, a
     biotic origin is more likely than abiotic synthesis.
      In addition, Popov (2023) questioned the age of the Volyn biota, which we postulated as
between approximately 1.5 and 1.7 Ga. He argues that the fossils could be Phanerozoic. We
will also outline our arguments for the minimum age of 1.5 Ga.

**1 Introduction**



We thank Head et al. (2023) for stimulating the discussion about the Volyn biota. They
question that these are fossils, instead argue that at least some of them are young
contaminants by plant hairs and pollen. This could have occurred during storage as what they
called 'museum dust' or during sampling. Furthermore, they question the biogenicity and
argue for an abiotic origin. We appreciate their comment, because this question of
contamination was not raised before, neither in our papers from 2017, 2022, and 2023, nor in
any of the previous publications about kerite who either described kerite as an abiogenic
material (Ginzburg et al., 1987; Luk'yanova et al., 1992; Yushkin 1996, 1998) or as fossilized
cyanobacteria (Gorlenko et al., 2000; Zhmur, 2003).
**2 Occurrence of kerite and sampling**
The following information is based on logbooks from the mine (VC, mine geologist in the area
since 1990) who also collected the material for our study together with PL. The samples of
kerite occur in situ underground in several, but not all shafts of the Volyn pegmatite district.
Within the large, miarolitic cavities ('chambers' in the original literature) kerite is also found
in the mineral matrix (feldspar, mica, clay minerals) on the floor of the pegmatite and is also
hanging from the walls or the ceiling. However, kerite in visible amounts is not preserved in
most chambers. It was either destroyed during cleaning and gemstone extraction, or it was
already collected. In those chambers which were explored by drilling, it was completely
destroyed by drilling fluids mixed with clay that covered the whole ground of the chambers.
Well-preserved large amounts of kerite were found only in new pockets opened by miners
underground without drilling. In January 2013 kerite was found (PL) in a 5 mm wide zone
around topaz crystals on the wall of a 15 m tall chamber in shaft 3. Kerite was observed
growing at the base of dark lilac black fluorite crystals, in larger fiber masses around large
topaz crystals, as larger fiber masses in clay along the lower walls and as large masses on well
crystallized feldspars, mica, quartz and topaz high on the walls in two chambers.
Early descriptions in the drilling logbooks mention in some cases that chambers were full
of kerite, up to 25 kg of kerite(!) in the rather small pegmatite body from shaft 3, which has
accesses to several pegmatite bodies (consistent with reports in the literature, e. g. Ginzburg
et al., 1987). Material from this shaft was distributed to museums in the former Soviet Union.
The chambers are now in a depth of up to 96 m, some were mined in open pits, but the
crystallization depth of the pegmatites was at a depth corresponding to 2-3 kbar. Thus,
significant uplift had occurred since intrusion at 1.76 Ga, but there is no indication from the
geological literature of the area that the chambers were directly on or beneath the surface
and buried again later. Therefore, contamination within the chambers by plant roots going
down to 96 m is less likely. In any case, we have no doubt that kerite is part of the deep
biosphere. Most trichomes (plant hairs) are known from plants on the surface, not from deep
biosphere.
Samples kerite 1 to kerite 7 were sampled underground by PL and VC, put into firmly
closed plastic sample bags (double ones with label in outer one), transported first to
Luxemburg and then sent to Berlin. There was no need to separate kerite from the rocks and
from the soil, the material could be picked up. Sample bags were opened only in the electron
microscopy laboratory of TU Berlin, which is a special building for electron microscopy with
the appropriate arrangements to prevent contamination by dust. All rooms are equipped with
airlocks for climatization and in addition water-cooled ceilings minimizes airstream and dust
movement in the rooms. Samples were prepared in an exhaust hood. Of course, we cannot
completely rule out that some objects are contaminants, but the overwhelming majority of
objects on the aluminum sample holders for scanning electron microscopy (SEM) are original



as recovered from underground. The only kerite sample, which could have been contaminated
in a museum is our sample 'kerite 0'.
96        The beryl crystal sample V2008 was collected from the mine tailings in 2008 by GF, stored
at TU Berlin in a common wooden rock cabinet. For this sample, contamination on the mine
tailings or later is possible.
99        The breccia with the beryl pseudomorph was also collected from the mine tailings in 2008
by GF, stored at TU Berlin in a common wooden rock cabinet, and consolidated with epoxy for
preparation of thin sections and polished blocks for the Ar-Ar-determination of muscovite.
**3 Composition and structure of kerite**
**3.1 Organic matter in the beryl pseudomorph**
We start the discussion with the OM in the pseudomorph. For this, a later contamination can
safely be excluded, as it was discovered in thin sections. It is closely surrounded and
intergrown with macroscopically black, in thin section brown, C-H-bearing opal (Franz et al.
2017; see their fig. 6). The chemical composition of the OM is characterized by a high amount
of Zr, Y, Sc, and REE. These high fieldstrength elements (HFSE) are positively correlated with
O, and increasing O contents are correlated with decreasing C contents. The N content is
between 2 and 4 at%, much lower than the original kerite (see their fig. 7), which has near 8-
9 at% (Ginzburg et al., 1987; Yushkin, 1996). Mobilization of HFSE is possible with a F-rich fluid
(Loges et al. 2023), and a high F-content in the system is likely because the pegmatites
themselves belong to the Nb-Y-F-type and contain a high amount of topaz. In addition, the
muscovite in the breccia is F-rich, and fluorite is a common mineral associated with kerite (see
below). For further details such as transmission electron microscopy of the border zone of OM
to opal and about opal itself, the reader is referred to our original publication.
118       We postulated that the low N-content was caused by decay of kerite, producing $NH_4$,
which was responsible for K-$NH_4$ exchange reactions in K-feldspar and in muscovite, forming
buddingtonite and tobelite. There is no doubt that before the formation of the breccia and
the pseudomorph, OM was present in the system. Buddingtonite is not a rare mineral in the
Volyn pegmatite field (Proshko, 1987) and the high activity ratio for $NH_4^+/K^+$ required to
transform K-feldspar into buddingtonite (Mäder et al., 1996) indicates a large amount of
decayed OM. This is not consistent with Head et al.'s concern that the OM in the pegmatite
field is late-stage contamination. Also, the chemical composition of the OM is completely
incompatible with anything like museum dust or plant hairs.
**3.2 Fossil or non-fossilized OM**
Head et al. (2023) question the fossil character of kerite. Here we want to summarize the
presented information about the metamorphic, mature character of kerite.
131       After the occurrence of OM in the pegmatitic environment, the temperatures had
reached again approximately 300 °C (Franz et al. 2017). This estimate is based on the phase
equilibria with bertrandite and muscovite in the pseudomorph. Furthermore, within beryl we
observed fluid inclusions with C-H, which occur on cracks sealed by secondary beryl (Voszniak
et al., 2012 ). This implies that temperatures were above the lower thermal stability of beryl,
which is at low pressure near 300 °C (Barton and Young, 2002). These temperatures are
consistent with our observation on decomposition of chitin to chitosan described in detail in
Franz et al. (2023a), see below the discussion about FTIR data.
139       All kerite samples were investigated by open-system pyrolysis. They do not differ
significantly, and all spectra show characteristics of mature to very mature OM (figure 13 in
Franz et al. 2022, and in supplement). This excludes young contamination by plant hairs.



Similarly, the light microscopic investigations in cross sections with white and UV light show
clear indications by different reflectivity and fluorescence, not consistent with young OM. We
described brittle behavior of kerite, also not compatible with young unmetamorphosed OM.
Brittle behavior was also noted by Yushkin (1996). Luk'ynaova et al. (1982) described X-ray
diffraction investigations with a diffuse peak at 8° Theta indicating OM with some graphite-
like sheets.
Head et al. (2023) refer to mineralized trichomes (Mustafa et al. 2017, 2018; Ensikat et al.
2017) and take this as an argument against fossilization. These plant hairs are biomineralized
with Ca-carbonate, Ca-phosphate and silica, especially at the tip of the trichomes. This
biomineralization is quite different from what we interpreted as fossilized and mineralized
rims of the Volyn kerite. We wrote that the most conspicuous feature is the common
occurrence of Si-Al-O, interpreted as Al-silicates. In the quoted investigations Al was never
observed. Furthermore, Ca-phosphate was observed in kerite only at some places at nano-
sized crystals (see e.g. figure 11 in Franz et al. 2022), at variance with a continuous
biomineralization on the tips. Kerite is completely surrounded by a mineralized rim, whereas
trichomes are only mineralized at their tips. All different kerite morphologies are mineralized
in the same way.
Concerning the analytical procedure applied by us, there is a misunderstanding in Head
et al.'s (2023) comment. On line 146 to 149 they wrote: "Had Franz et al. (2023) used EDX in
addition to applying EDAX EDS to selected cross sections, they would have been easily able to
determine the elemental distribution for all specimens they imaged using SEM which could
have assisted in discriminating extant contaminations from fossil material." For our element
mapping we used wave length dispersive (WDS) analysis with the electron microprobe (EMP),
which is much more sensitive than energy dispersive systems (EDS) such as EDAX. We have
shown several element distribution maps of different morphologies in Franz et al. (2022), and
since all show generally identical features with an Al-Si-Ca rim structure and an internal
structure with characteristic N-O-S distribution, we can safely exclude biomineralization, but
instead mineralization due to a fossilization process.
### 3.3 EDS (with SEM)
All spectra of kerite objects show a high amount of oxygen. This excludes fresh organisms but
indicates again (highly) mature OM. Minerals on the surface of filamentous kerite (Fig. 1a-d),
of bulbous kerite (Fig. 1e), and of the spherical object, interpreted by Head et al. (2023) as a
pollen, are mostly Al-silicates, some with K, Na, and Ca. The flaky shape of the minerals
indicates clay minerals, one needle-shaped crystal is a Ti-oxide, possibly rutile.



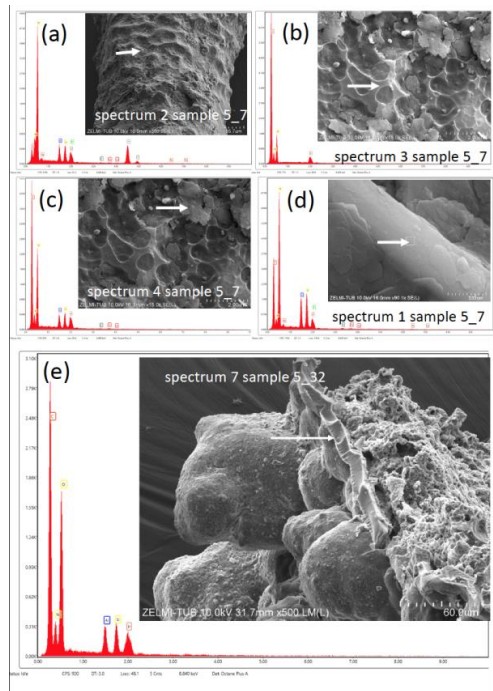

**Figure 1.** EDS spectra obtained with the SEM of filamentous (a, b, c, d) and bulbous (e) kerite
objects. (a) Needle-shaped small object with high Ti-O contents (arrow; interpreted as rutile)
next to Al-silicates with minor amounts of Na, K, and Ca. (b) Spectrum of clear surface (arrow)
of kerite, showing only the kerite composition of C-N-O. (c) Spectrum of platy mineral grains
(arrow) with Al-Si and small contents of K and Fe, interpreted as a clay mineral. (e) Base
(arrow) of bulbous kerite, with high amounts of Al-Si. Samples are iridium-coated.





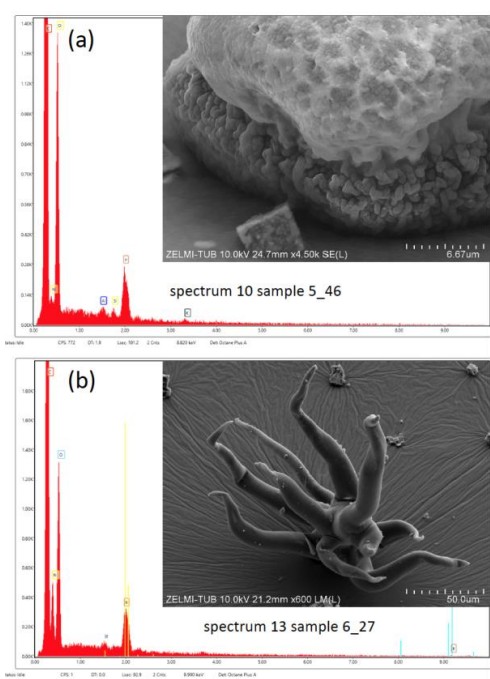

**Figure 2.** EDS spectra of a spherical object (a) and a filamentous object (b). The spherical object
shows Al-Si-K peaks, which can be interpreted as illite, whereas the filamentous object shows
only the typical composition of kerite with C-N-O; both samples are iridium-coated.

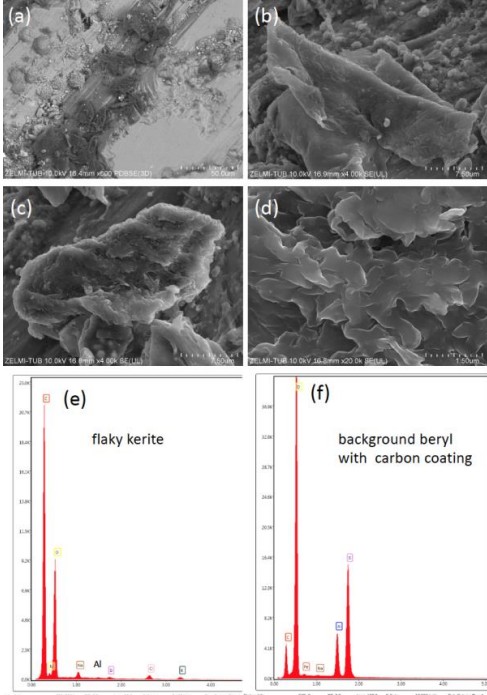




**Figure 3.** EDS data of flaky kerite, observed on sample V2008, a beryl crystal. (a, b, c, d) show the structure of kerite, in (a) with combined BSE detector for element contrast. The dark contrast compared to background beryl and other minerals indicates low average atomic number. (e) is the corresponding EDS spectrum with clear indication for Si, Al, Na, K, and Cl, next to C-N-O of kerite. (f) is EDS spectrum of beryl; note the low C-peak caused by carbon coating, compared to the large C-peak of kerite.

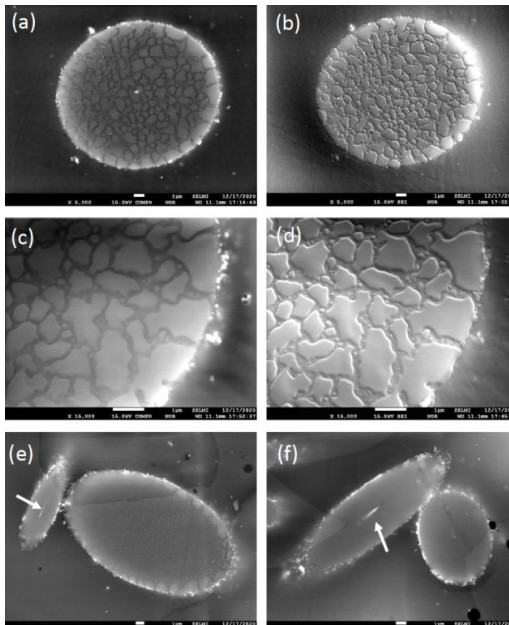

**Figure 4.** BSE (a, c, d, e) and SE (b, d) of cross sections of filamentous kerite, embedded in epoxy. Note the discontinuous rim of high contrast indicating mineralized parts, and within the channel (e, f) also with high contrast (arrows). The mosaic pattern with different contrast in BSE (a, c) is seen in SE images (b, d) as slightly lower areas of approximately 200 nm width.

The EDS spectrum of the object, interpreted as pollen by Head et al. (2023), also shows the presence of Al and Si, together with the typical C-N-O peaks (Fig. 2). The EDS spectrum of the object, interpreted by Head et al. (2023) as trichome 'museum dust' (Fig. 3) shows no Al-Si, but the C-N-O ratios are very similar to those of the mineralized filaments, and therefore we have no doubts that this is also fossilized OM.

### 3.4 EMPA data

In BSE images of cross sections of filamentous kerite we see a discontinuous mineralized rim (Fig. 4). In combination with the element mappings (see images in figures 8 to 11 in Franz et al. 2022, and figures S6, S7 in the supplement to Franz et al. 2022), we can safely conclude that the mineralized rim consists dominantly of Al-silicates. Some other minerals such as Ca-phosphate or silica occur only in isolated spots and do not cover the whole rim. Over a distance of approximately 1 µm the filament shows a higher contrast rim in BSE images, indicating a higher average atomic number, consistent with our interpretation that this is caused by a mineralized, impregnated rim of dominantly Al-silicates. In the internal structure of the filament, a mosaic patter can be observed with approximately 200 nm wide channels, also



indicated by different element contrast (Fig. 4a, b). In SE images (Fig. 4c, d) a slightly lower
position of the channels is seen, caused by different behavior during polishing. This internal
structure is compatible with fossilized material, not with fresh cells of trichomes.
**3.5 TEM data**
In addition to the transmission electron microscope (TEM) investigations we presented in
Franz et al. (2017), we cut a new focused ion beam (FIB) foil from a filamentous object (Fig.
5). The foil covers the embedding material epoxy (characterized by typical Cl-content), the
approximately 500 nm wide rim and kerite (with dominantly C-O and N). The rim consists of a
mixture of different minerals, which can be distinguished by different contrast in the HAADF
images. EDAX spectra indicate dominantly Al-silicates with minor amounts of K, Ca, and Fe,
and a Fe-Ca-phosphate. This is different from the type of biomineralization in trichomes,
shown by Mustafa et al. (2017, 2018) and Ensikat et al. (2017).

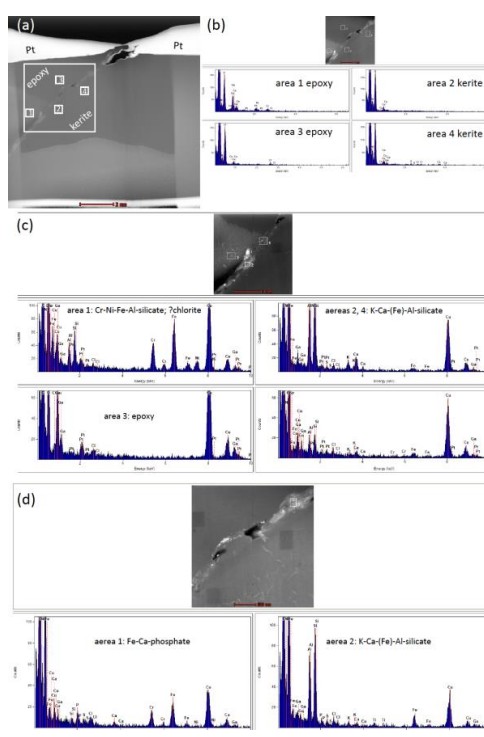

**Figure 5.** Analytical EDAX-TEM results on a FIB from the rim of a filamentous kerite object.
Note for all spectra that Ga-peaks are due to the Ga ion cutting, Cu peaks originate from the
copper grid, and Pt from the platinum holder. (a) Overview of the FIB foil; white frame
indicates position of (b) high-angular annular dark-field (HAADF) image and EDAX spectra of
kerite and embedding material epoxy. (c) Detail of (b) with EDAX spectra of three inclusions,
interpreted as possibly chlorite and a complex Al-silicate, possibly a clay mineral. (d) Detail of
(b) with EDAX spectra of two inclusions, a Fe-Ca-phosphate and a complex Al-silicate.
**3.6 IR spectra**
Head et al. (2023) criticize our IR spectra and argue that we should have used modern fungal
chitin standards for comparison and a more detailed comparison with sub-fossil and fossil
fungi. Since we knew that the Volyn biota experienced temperatures near 300 °C, comparison



with modern fungi did not seem appropriate to us. Instead, we followed the procedure by
Loron et al. 2019) and the thermal degradation studies of chitosan (Wanjun et al., 2005;
Zawadzki and Kaczmarek, 2010; Vasilev et al., 2019). These are clearly consistent with our
conclusion that chitosan is a constituent of the kerite material.

**4 Comparison of kerite morphology**

Head et al. (2023) present evidence for strong similarity of one object of our sample collection
with pollen of an extinct conifer. The similarity is indeed striking, but we want to stress
important differences (Fig. 6):

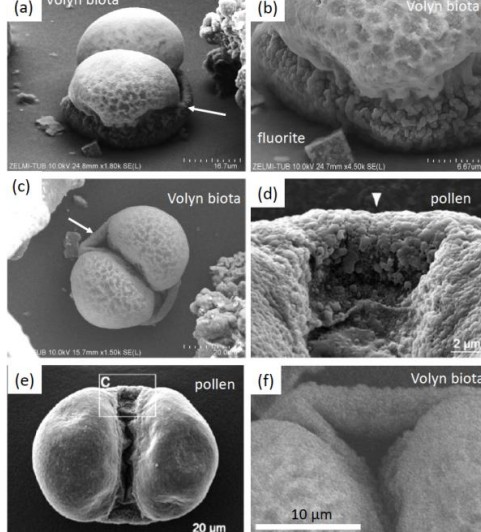

**Figure 6.** SEM images for direct comparison of kerite object from Volyn biota (a, b, c, f) and
*Pinus* pollen (d, e) from Head et al. (2023). Note that the kerite object is sitting firmly on base
consisting of OM (a, b), whereas pollen are free objects. The surface of kerite is characterized
by dents (b), whereas the pollen shows a microrogulate surface (d). What is described from
pollen as air sacs (d, e) sits on a similar height as the pollen grain itself (d), whereas what we
described as a sheath comes from the base of the kerite object (arrows in a and c). This sheath
shows some inward folding (f), which is not seen in the air sac of the pollen.

The kerite object is sitting firmly on a base consisting of OM (Fig. 6a, b), whereas pollen
are free objects. The surface of kerite is characterized by dents (Fig. 6b), whereas the pollen
shows a microrogulate surface (Fig. 6d). What is described from pollen as air sacs (Fig. 6d, e)
sits on a similar height as the pollen grain itself, whereas what we described as a sheath comes
from the base of the kerite object (arrows in Fig. 6a, c). This sheath shows some inward folding
(Fig. 6f), which is not seen in the air sac of the pollen.
The other object, which Head et al. (2023) interpret as a plant hair (figure 3 j, k, l; in Franz
et al. 2023a) due to the similarity to 'museum dust', also sits firmly on a base. If such a delicate
object like unfossilized trichome was transported down into the chamber (where it was
sampled), it is difficult to imagine that it survived the transport.
Head et al. (2023) restrict their criticism to these two objects but do not mention the fact
that the large majority of objects we presented has a different morphology, with filaments up
to the mm-size, bulbous objects, objects with irregular shape etc. None of these objects is



similar to trichomes. Also, they do not mention the internal structure with a channel, which
we documented in detail (figure 11 in Franz et al., 2023a), and which is obvious also in BSE
images (Fig. BSE e, f). They also do not mention the presence of Bi(Te,S) biomineralization,
which we documented (figure 10 in Franz et al., 2023). To the best of our knowledge, this type
of biomineralization was not observed in trichomes.

**5 Age of the fossils**

Popov (2023) questioned the minimum age of the organic matter, which we proposed as 1.5
Ga, based on the Ar-Ar laser ablation data (Franz et al., 2022a) of muscovite in a pseudomorph
after beryl. He proposed a sequence of events, starting with the intrusion of the granites and
the pegmatites at approximately 1.76 Ga (Shumlyanskyy et al., 2017, 2021), cooling and
pseudomorph formation due to a hydrothermal event at 1.5 Ga, then again cooling,
introduction of organic matter, then a second hydrothermal event, which converted
muscovite into tobelite and K-feldspar into buddingtonite. The age of the second event could
have been early Phanerozoic, based on our data (Franz et al., 2022b) of dating attempts of the
kerite itself, which produced in Popov's (2023) wording an isochrone of 493±98 (1s) Ma, but
which we considered only as a reference line due to the large uncertainty. In this sequence of
events the breccia formation is missing, but this event is important: It fractionated feldspar
and quartz into cm-sized, irregular pieces, including a large piece of pegmatitic beryl. This
event must have occurred before the pseudomorph formation, because the delicate
pseudomorph, consisting of a rather loose framework of muscovite and bertrandite would not
have survived the brecciation. But the breccia is cemented by black opal (pigmented by
hydrocarbons), and OM must have been present before precipitation of opal. Therefore, the
sequence of events after the intrusion must have been: Presence of organic matter,
brecciation, pseudomorph formation at 1.5 Ga in one event, first with muscovite formation,
then during decay of the kerite and production of $NH_4^+$ tobelite and buddingtonite (including
formation of secondary, C-H bearing fluid inclusions in low-T beryl), then further cooling. It
was made clear in our text that "…the fluid composition changed during the pseudomorph
formation, starting with F-dominated K-rich fluids producing pure F-muscovite, followed by
alternating $NH_4$-rich and K-rich compositions, producing oscillatory growth zones in
buddingtonite (Fig. 5e) and ending with a late K-rich fluid (producing some outer K-rich zones
in buddingtonite; Fig. 5d)." This is the same conclusion as in our first analysis of the
pseudomorph's texture (Franz et al., 2017) and clear from the summary figure 13, illustrating
the sequence of processed in one single geological event. We feel misinterpreted by Popov
(2023), who wrote that in our second study we had changed our mind.

313        There might have been additional hydrothermal events since 1.5 Ga, caused e. g. by the
Neoproterozoic Volyn Large Igneous Province at approximately 600 Ma or later Devonian
rifting of the Prypyat aulacogen (Shumlyanskyy et al., 2016), but none of these events is
documented up to now in the pegmatites of the Volyn field. We fully agree with Popov (2023)
that the late-stage development of pegmatites including later overprinting by hydrothermal
events may point to a protracted history. However, for the Volyn locality, the late-stage
development is documented in Lazarenko et al. (1973) and in a study of dissolution of Volyn
beryl crystals with the formation of typical and diagnostic etching (Franz et al., 2023b).

**6 Origin of kerite - biotic or abiotic**

Head et al. (2023) conclude their discussion with "We have doubts whether any of the in-situ
Volyn 'biota' is organic in origin", based on references to the low $\delta^{13}$C values obtained via
experiments with Fischer-Tropsch-type synthesis under hydrothermal conditions in the



presence of metallic Fe. From a starting composition with an assumed value for $\delta^{13}$C of -20 ‰,
different organic compounds were obtained with a rather uniform composition of -50 ‰
(McCollum and Seewald, 2006). Abiotic synthesis of nitrogen-bearing organic carbon species,
such as amino acids, is thermodynamically favored by molecular $H_2$, which is produced by
serpentinization of Fe-rich mantle-derived rocks (Ménez et al., 2018).
Source for abiotic synthesis in Volyn should be the mantle with a uniform $\delta^{13}$C value of
-5 ‰ (Marty et al., 2013, and references therein), because the Korosten pluton is comprised
mainly of mantle-derived granitic, gabbroic, and anorthositic rocks (Shumlyanskyy et al., 2017,
2021). Assuming a similar fractionation of -30 ‰ for a source with $\delta^{13}$C of -5 ‰, a composition
of abiotic kerite should have values of -35 ‰, but many kerite bulk samples have much lower
values between -40‰ and -48‰. According to the model of abiotic origin, mantle-derived
fluids should also be the source for nitrogen. The N-isotopic signature of the mantle scatters
from -25 ‰ to +15, with most values around -5±3 ‰ Cartigny (2005), therefore a mantle
source is less likely for the Volyn locale, with positive $\delta^{15}$N values up to 10 ‰ throughout. An
alternative source might be the country rocks of the Korosten pluton, but this would require
a large amount of C- and N-rich fluids, and there is no geological evidence for such fluid-rock
interactions. They should have left their signature also within the granites, which are the hosts
of the pegmatites. The presence of Fe-rich minerals as catalyst for the production of abiotic
carbonaceous material in serpentinites (Nan et al., 2021) is not a good analogue for the
granitic environment, in which Fe-rich minerals are generally scarce. Yushkin (1996) presented
analyses of different proteins in kerite and used it as an argument that abiotic synthesis is
possible. However, he starts from the assumption of abiotic origin and did not consider the
possibility of fossil material. The large amounts of kerite of several kg recovered from the mine
(Ginzburg et al., 1987) is in our view more consistent with biomass accumulation; what has
been described as abiotic formation of carbonaceous material was observed in small amounts
in thin section only (e. g. Nan et al., 2021; Ménez et al., 2018).

**Summary and open questions**
Although the morphology of two objects, selected by Head et al. (2023) show a striking
similarity to recent organisms, the combination of all observations is much more in favor of
fossil organisms: The occurrence in the mine as part of the deep biosphere; a large variety of
morphologically different objects, which however have all the same type of rim
mineralization; their brittle behavior; the internal structure with a channel in the filaments;
the presence of biomineral inclusions of Bi(Te,S).
We are aware that our single age determination of 1.5 Ga for the hydrothermal
overprinting of the pegmatites should be verified or falsified by ages on different minerals
and/or different isotope systems. If more and better data will be available, we are happy to
change the interpretation, but with the current available data the presented interpretation
seems to be the best one. Fluid inclusion studies might further help to clarify the origin of
kerite (Vozniak et al., 2012, and references therein; Kalyuzhnyi et al., 1971). Liu et al. (2022)
observed whewellite ($CaC_2O_4 \cdot H_2O$) in $CO_2$-$N_2$-$CH_4$-vapor of fluid inclusions in topaz, thought as
a product of oxidation of organic material with an alkaline fluid. In-situ determination of C-
and N-isotopes, and possibly also other stable isotopes (e. g. O, S) might also help to further
clarify the type of organisms, their internal structure, and their origin.




*Data availability.* All data are as figures in the text or in the cited references.

*Supplement.* There is no supplement to this article.

Author contributions. GF concept, figures, and writing; VK writing, FTIR; VC and PL information
about the sampling and occurrence, US writing, stable isotopes.

*Competing interests.* The authors declared that they have no competing interests.

*Acknowledgements*. We thank Anja Schreiber and Richard Wirth for permission to use
unpublished TEM data.

*Financial support.* VK acknowledges funding by Alexander von Humboldt foundation.

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
