# Peer review of "Reply to Comment on Franz et al. (2023): A reinterpretation of the 1.5 billion year old Volyn"

_EGUsphere, 2024_

## Author Response (AR1)

Letter to the AE

Dear Cindy,

Thank you very much for handling our manuscript!

As the reviewers and you suggested, we added a short paragraph about kerite and its origin in the Introduction.

Concerning the additional methods, you mentioned in your AE comments
i) we added the sentence: *A more detailed description of isotopic composition of the kerite organic matter might be possible by in-situ methods. Such methods are currently not available for us, but we will explore the possibility for cooperation with other laboratories.*
In the section Summary and open questions, we also added a sentence: *Further studies on the molecular composition, i. e. certain biomarkers, will help to characterize kerite in more detail and give information about the type of organisms, which requires, however, more material. This is under the current situation in Ukraine not available.* (We had already sent one sample, for which we had enough material, to Christine Heim (Univ. Köln, Germany). This sample gave some results, but not totally satisfactory.)

ii) determining the origin of kerite: We already mentioned in the last section fluid inclusion studies. I don't see any other methods (except those which we will mention for the other points).
iii) age of the kerite; we added: *We are currently working on Rb-Sr data with the laser-ablation system from the same sample, which was determined by Ar-Ar and which can be applied to both minerals, muscovite and feldspar. Additional sample material with white mica and feldspar is also available and will be studied.*
iv) the confirmed presence of chitosan; we don't see the necessity to go further than what is written in the current version of the reply.
We also added the suggested sentence: *Fischer-Tropsch Type (FTT) process is unlikely in the current geological setting of an Fe-poor granite-pegmatite system.*

Please let us know if these changes are sufficient and ok, but of course we are happy to introduce further comments or make changes.

Best regards, on behalf of the authors
Gerhard Franz